# Liposomal Inhalation after Tracheostomy—A Randomized Controlled Trial

**DOI:** 10.3390/jcm10153312

**Published:** 2021-07-27

**Authors:** Benedikt Hofauer, Ulrich Straßen, Adam Chaker, Beate Schossow, Magdalena Wirth, Markus Wirth, Murat Bas, Andreas Knopf

**Affiliations:** 1Otorhinolaryngology/Head and Neck Surgery, Klinikum Rechts der Isar, Technical University of Munich, 81675 Munich, Germany; b.hofauer@tum.de (B.H.); u.strassen@tum.de (U.S.); a.chaker@tum.de (A.C.); magdalena.wirth@tum.de (M.W.); markus.wirth@tum.de (M.W.); m.bas.hno@gmail.com (M.B.); 2Study Center Munich, Klinikum Rechts der Isar, Technical University of Munich, 81675 Munich, Germany; beate.schossow@mri.tum.de; 3Otorhinolaryngology/Head and Neck Surgery, University Medical Center Freiburg, University of Freiburg, 79106 Freiburg, Germany

**Keywords:** tracheostomy, inhalation, liposomes, physiological saline solution, interleukin

## Abstract

Background: Tracheostomy is a common procedure in critical care. The aim of this study was to evaluate the application of a liposomal inhalation compared to standard physiologic saline (SPS) inhalation on basis of objective and subjective parameters of airway inflammation. Methods: We evaluated in this two-armed, double-blinded and randomized control group study the effect of liposomal compared with SPS inhalation in newly tracheotomized patients. The primary endpoint was defined as trend of tracheobronchial IL-6 secretion at day 1 compared to day 10. Further objective and subjective parameter were evaluated. Results: Fifty patients were randomized in each arm. Tracheal IL-6 levels decreased significantly only after liposomal inhalation. Both inhalative agents seem to have an effect on the respiratory impairment after tracheostomy. Subjective patient impairment was reduced significantly from day 1 to day 10 after tracheostomy with liposomal inhalation. Conclusions: Liposomal inhalation demonstrated an advantage over SPS inhalation in newly tracheotomized patients.

## 1. Introduction

Tracheostomy is one of the most common procedures in patients in the critical care setting and can either be performed as an open surgical procedure or as percutaneous dilatative procedure [1,2]. Various factors, such as the availability of bed-side ventilators for long-term domiciliary care and the increase of patients with end-stage respiratory disease or with permanent damage after a neurological, neurosurgical or traumatic event, lead to an increase in the numbers of patients with temporary or permanent tracheostomy [3,4,5]. Both in the USA and across Europe, 7–16% of critical care admissions are managed with a tracheostomy [6]. With more than 100,000 tracheostomies performed annually in the USA, caring for these patients (critical care, rehabilitation, chronic care, home care) has become an important issue [7]. Additionally, around 25% of patients with malignancies in the area of the upper aerodigestive tract receive a tracheostomy during the therapeutic management of the disease [8]. 

Despite its multiple benefits in the clinical management, a tracheostomy interferes with relevant function of the upper airways. The physiological conditioning of the inhaled air (humidification, filtration, balance of temperature) and retention of a significant amount of water from the exhaled air ceases to exist due to the bypass of the upper respiratory tract including the sinonasal and oropharyngeal tract. The loss of the sinonasal function leads to pathological changes of the lower airways including an impaired function of the cilia and a consequent loss of mucociliary transport. Spontaneously breathing patients often complain of coughing, excessive production of secretion and crusting especially in the acute phase after the tracheostomy [9]. This results in repeated cleaning and suction of the lower airways, a significant discomfort for the patients, and increases the risk of lower respiratory tract infections and airway obstructions. It has been shown that in patients with long-term tracheostomy the airways above the carina cannot be kept in good order, which manifest in a chronic inflammation of the trachea and bronchi and airway remodeling [9,10,11,12,13,14]. 

Different possibilities are available to replace the function of the sinonasal tract to conditionate the inhaled air. A common method is the application of heat and moist exchangers (HME), passive humidifiers which retain heat and humidity within the tracheobronchial system [15]. Especially in the early postoperative phase, heated and non-heated nebulizers are widely recommended to facilitate the omission of the upper airways. However, despite the common use of comparative data, the objective and subjective effects of different inhalative agents are missing and no general valid recommendations exist at present. Therefore, the aim of this study is to evaluate the effects of application of a liposomal inhalation, which contains phospholipids and enables, in addition to the humidification of the respiratory tract, the supplementation of important constituents and restoration of the surfactant film. The hypothesis of this study is, that a liposomal inhalation when compared to standard physiologic saline inhalation, has a beneficial effect on objective and subjective parameters of airway inflammation. 

## 2. Study Design and Methods

This two-armed, double-blinded and randomized control group study was conducted at the Department of Otorhinolaryngology/Head and Neck Surgery at the Klinikum rechts der Isar, Technical University of Munich, Germany. The study protocol was in accordance with the Declaration of Helsinki. The Institutional Review Board of the Medical Faculty, Technical University Munich, reviewed and approved the protocol (41/14) and the study was registered on Clinical Trials (NCT02157129). The study was monitored by an independent study center (Munich Study Center). Written informed consent was obtained from all participants prior to the begin of the intervention. 

### 2.1. Study Population

Adult patients were included prior to an elective tracheostomy or within ≤24 h after tracheostomy. Patients with known allergies for ingredients of the inhalation, >24 h after tracheostomy, acute or imminent sepsis, existing bronchopulmonary inflammation, immunosuppressive therapy, poorly adjusted pulmonary disease and chronic respiratory insufficiency were excluded. 

### 2.2. Study Protocol

After inclusion, patients were randomized 1:1 into two groups: group 1 received a liposomal inhalation (LipoAerosol^®^, Optima Medical Swiss AG, Zug, Switzerland), containing phospholipids, which is a major component of surfactant, and group 2 received an inhalation with physiologic saline inhalation, the current gold standard, from the first to the tenth day after surgery, five times at 30 min per day (PARI NaCl Inhalation Solution, Starnberg, Germany). The liposomal inhalation contains liposomes, i.e., phospholipid bilayer vesicles, which shape the main constituents of surfactant film, which covers the air/liquid interface on the airways from the lower to the upper respiratory system [16]. LipoAerosol^®^ is a medical device in accordance with Medical Device Directive 93/42/ECC, which obtained CE-marking in 2012 as the first commercially available liposomal inhalation solution and is based on a physiological saline solution with the addition of phospholipid-liposomes made of highly purified lecithin. The further study flow is depicted in Figure 1. Two physicians were involved in every included patient. The first physician was responsible for the randomization and the preparation of the solution for the inhalation. The study inhalation was prepared in an opaque container to prevent another person from distinguishing the different inhalative agents from each other. Both the patient and the second physician, who performed the examinations during the study period, were blinded towards the used inhalative agent. In all patients in both groups heat and moist exchanger (HME) filters were used. The inhalation was prepared by a physician, who was not involved in the further examinations. All included patients were inpatient for the entire study period. 

### 2.3. Outcome Parameter

On day 1 (baseline), day 3 and day 10 after tracheostomy various objective and subjective parameters were evaluated. The primary endpoint was a change in the tracheal interleukin 6 (IL-6) levels at day 10 after tracheostomy. As part of routine suction maneuvers, the tracheal secretion was collected in a secretion trap, immediately cooled on ice, and stored at −20 °C until evaluation. The in the tracheal secretion which found inflammatory cytokine IL-6 is measured quantitatively (Roche Diagnostics GmbH, Unterhaching, Germany). The secondary endpoint was a change in the respiratory impairment at day 10 after tracheostomy, which was evaluated with a medical scoring system, which included the frequency of suction maneuver (0 points: none, 1 point 5–10 x/d, 2 points: >10–20 x/d, 3 points >20 x/d), an bronchoscopical assessment of the tracheobronchial redness (0 point: none, 1 point: peristomal, 2 points: tracheal, 3 points: trachea-bronchial) and an assessment of the thickness and consistence of the mucous congestion (0 point: none, 1 point: fluent, 2 points: tenacious, 3 points: barky). Other endpoints were changes in inflammatory serological and tracheal secretion parameters at day 3 and 10 after tracheostomy (serological: C reactive protein (CRP), white blood cell count (WBCC) and changes in the subjective overall impairment at day 3 and 10 after tracheostomy (evaluation of the severity of coughing frequency, breathlessness, thickness and consistence of mucous congestion, color and consistency of tracheal secretion with visual analogue scales). 

### 2.4. Statistical Analysis

Statistical analysis was done using version 25.0 of the Statistical Package for Social Sciences software (SPSS, Chicago, IL, USA). Descriptive data are reported as mean ± standard deviation, if not otherwise stated, and were compared between both groups. Paired *t* tests were used for continuous variables and χ^2^ tests for categorical variables. When necessary, Mann–Whitney U tests were used. *p*-Values of less than 0.05 were considered as statistically significant. A calculated number of *n* = 100 patients (standard deviation = 7.5; α = 5%; power = 80%; difference in means = 0.42) per examination arm is assumed. An interim analysis after 100 randomized cases was declared and a consequent termination of the study in case of significant results regarding the primary endpoint. Since no comparative values were available for the development of the IL-6 values in the tracheal secretion, we referred to the assumed reduction in the number of points in the medical scoring system when estimating the number of cases. The randomization was organized with consecutive envelopes, which were prepared by a statistician (Dr. rer. nat. Victoria Kehl, Munich Study Center/IMSE) who was independent of the subsequent evaluation. The envelopes were opened when the patient signed their consent form to participate in the study.

The randomization envelopes are made on the basis of a randomization list. The randomization list for this study was generated using nQuery Advisor v. 7.0 (Statistical Solutions Ltd., Cork, Ireland). The allocation to the two therapy groups was 1:1 and in blocks (variable block length).

## 3. Results

### 3.1. Study Population 

The study populations consisted of 100 patients of whom 50 were randomized to receive a liposomal inhalation in the postoperative phase after tracheostomy and 50 patients were randomized to receive an inhalation with physiologic saline. Patients were recruited between June 2016 and June 2017. The CONSORT flow chart is depicted in Figure 2. 

One patient in group 2 discontinued the blinded inhalation of physiologic saline due to concerns of possible side effects (subjective sensation of shortness of breath) but continued the same inhalation unblinded without further complaints and was therefore included in the further analysis. Due to significant results on the primary endpoint the inclusion of patients was terminated after the interim analysis of 100 randomized patients. Analysis of the characteristics of the two groups did not reveal differences regarding age, sex, comorbidities, underlying kind and location of diagnosis which lead to the tracheostomy, kind of therapy, alcohol consumption and nicotine abuse (Table 1). The most important diagnoses, which were the indication for the tracheostomy in the entire cohort, were malignant diseases (96%). The malignant diseases were mostly located at the level of the oropharynx (44%) or the level of the larynx (40%). In 83% of cases, the tracheostomy was performed during the surgical therapy of the primary malignoma. Treatment overall was well-tolerated and no adverse events were reported.

### 3.2. Effect on Tracheal IL-6 Level (Primary Endpoint)

Patients treated with liposomal inhalation revealed a substantial decrease in tracheal IL-6 at day 10 (−71.32% in comparison to −8.23% in patients treated with physiological saline inhalation, *p* = 0.001). The baseline IL-6 levels at the first day after tracheostomy with levels of 31.975.81 pg/mL (±41,771.89) in group 1 and levels of 21,093.45 pg/mL (±25,280.55) in group 2 did not differ significantly (Table 2 and Figure 3). While in group 1 after liposomal inhalation the tracheal IL-6 level decreased to a level of 9491.67 pg/mL (±15,809.79, *p* < 0.001) ten days after tracheostomy, the tracheal IL-6 level in group 2 did not change significantly (19,358.50 pg/mL ± 36,130.19, *p* = 0.637). Therefore, the difference between IL-6 levels at day one and day ten after tracheostomy was higher in group 1 (liposomal inhalation, 22,484.14 ± 35,172.53 pg/mL) compared to group 2 (physiologic saline inhalation, 1734.94 ± 26,311.94 pg/mL, *p* = 0.001). 

### 3.3. Effect on Respiratory Impairment (Secondary Endpoint)

The evaluation of the respiratory impairment included the frequency of suction maneuver, a bronchoscopical assessment of the tracheobronchial system and of the mucous congestion. The baseline score of 3.2 ± 1.2 in group 1 and 3.1 ± 1.2 in group 2 did not differ significantly (*p* = 0.806) and were reduced in both groups ten days after tracheostomy (1.8 ± 1.6 and 2.4 ± 1.4, respectively, both *p* < 0.001, Table 3). Both inhalative agents seem to have an effect on respiratory impairment after tracheostomy; however, the effect of a liposomal inhalation was more pronounced, resulting in a difference between day one and day ten of 1.4 ± 1.8 compared to 0.7 ± 1.5 after inhalation with physiologic saline solution (*p* = 0.040).

### 3.4. Effect on Further Inflammatory Parameter (Other Endpoint)

The baseline levels of the WBCC and CRP are depicted in Table 4. During the postoperative course both levels decreased significantly in both groups (WBCC in both groups, *p* < 0.001; CRP in group 1, *p* < 0.001; CRP in group 2, *p* = 0.004) without any differences between the two groups. The decrease of the serological inflammatory parameter did not correlate with the development of the tracheal inflammatory parameter. 

### 3.5. Effect on Subjective Overall Impairment

The subjective overall impairment, consisting of the parameter coughing frequency, breathlessness, mucous congestion, color and consistency of tracheal secretion evaluated with visual analogue scales, was reduced significantly from day one to day ten after tracheostomy in group 1 (*p* < 0.001) but not in group 2 (*p* = 0.800, Table 5). Consequently, the difference between the score at day one and day ten was larger in group 1 compared to group 2 (6.93 ± 12.18 vs. 0.29 ± 8.00, *p* = 0.002). 

## 4. Discussion

The current study compared the parameter of airway inflammation in patients receiving either liposomal or physiologic saline inhalation after tracheostomy. Liposomal inhalation resulted in a decrease in proinflammatory tracheal IL-6 levels and provided greater improvement in respiratory impairment in the early postoperative course after tracheostomy compared to inhalation with physiologic saline solution. 

In previous studies on the mucociliary clearance in newly tracheotomized patients, Birk et al. demonstrated an impaired ciliary function in all the included patients [11]. The authors hypothesized that this observation could be explained by epithelial irritation caused by the tracheostomy. It has further been shown by Braun et al. that in a cohort of tracheotomized children, neutrophil numbers in bronchoalveolar lavage (BAL) fluid were elevated compared to controls, which indicates a bronchopulmonary inflammatory reaction, despite the lack of clinical symptoms in this cohort [17]. The higher frequency of positive bacterial culture in the BAL in these patients highlights the predisposition towards airway inflammation in tracheotomized patients [17]. According to the reported effect on the airways, elevated tracheal IL-6 levels, a proinflammatory cytokine and an important mediator of the acute phase response, have been measured in patients with tracheostomy compared to controls [18]. The influence of a substitution of surfactant on the limited mucociliary clearance has already been described for different types of applications in different situations [19,20]. In addition to a restoration of mucociliary clearance, a decrease of the IL-6 levels in tracheal fluid following surfactant therapy was observed, indicating its ability to modulate levels of inflammatory cytokines in the airways [21]. Taking these observations into account, sole humidification of the inhaled air in patients with tracheostomy appears insufficient to decrease inflammatory responses in the airways but rather (additional) strengthening of the surfactant appears necessary. 

The liposomal inhalation used in this study contains phospholipids and enables, in addition to the humidification of the respiratory tract, the supplementation of important constituents and restoration of the surfactant film. In both groups, the respiratory impairment could be reduced due to the moisturizing effect of both inhalations, but the level of proinflammatory IL-6 did only decrease in patients receiving a liposomal inhalation. It cannot be assumed that the tracheal IL-6 values decreased in accordance with the serological inflammatory parameters (white blood cell count and CRP), since these showed an equivalent decrease in both groups whereas the tracheal IL-6 values only decreased after liposomal inhalation in group 1. 

The two inhalations differed substantially in their effect on the subjective complaints of the patients (Table 5). While in group 2 (inhalation with physiologic saline solution, the current gold standard) no significant improvement was observed; patients with liposomal inhalation indicated their subjective impairment significantly improved, mainly due to effects on the subjective handicap due to the tracheostomy, on the coughing frequency and on the consistency of tracheal secretion (in agreement with previous reports) [22].

The key strength of the current study is the comparison of two different inhalation solutions in a prospective, randomized and double-blinded controlled fashion. The study is designed to provide robust evidence in a clinical setting, in which physiologic saline solution is used for inhalation in patients early after tracheostomy as standard inhalation lacking scientific rationale if superior inhalative agents are available. Our findings add to the literature, that an inhalation containing liposomes has a beneficial effect both on tracheal IL-6 levels and on further objective and subjective impairments in the postoperative phase after tracheostomy. Potential limitations of the current study are the short observation period of only the early postoperative phase of ten days. Therefore, it is not clear, for how long patients should use a liposomal inhalation and if or when a change to physiologic saline inhalation is possible. 

In conclusion, a liposomal inhalation demonstrated an advantage over physiologic saline inhalation in newly tracheotomized patients with regards to clinically relevant parameters. The level of tracheal proinflammatory IL-6 values as well as further objective and subjective parameters could be significantly improved in patients receiving liposomal inhalation. 

## Figures and Tables

**Figure 1 jcm-10-03312-f001:**
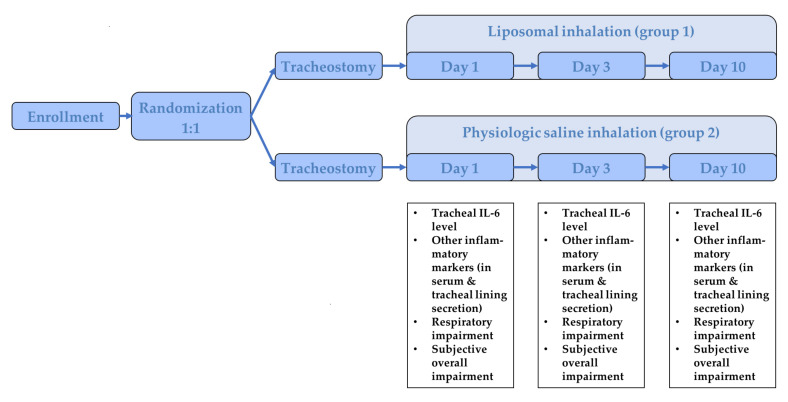
Illustration of the study flow.

**Figure 2 jcm-10-03312-f002:**
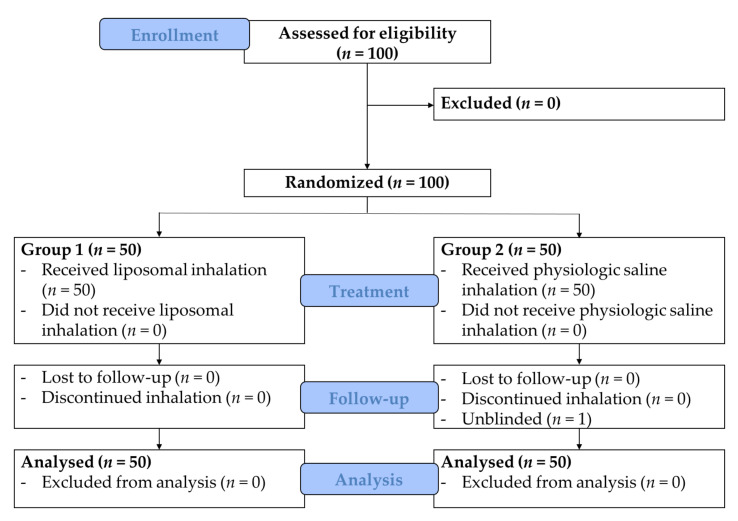
CONSORT flow chart.

**Figure 3 jcm-10-03312-f003:**
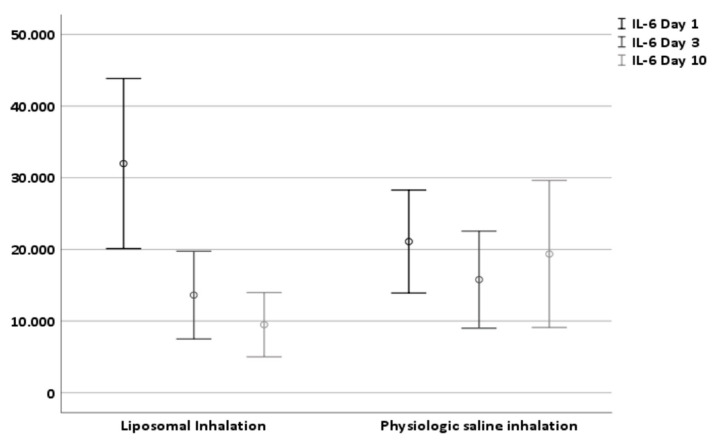
Effect of liposomal and physiologic saline solution inhalation on tracheal IL-6 level at day 1, day 3 and day 10 after tracheostomy.

**Table 1 jcm-10-03312-t001:** Comparison of baseline demographic and clinical characteristics of included and randomized patients.

Variables	Liposomal Inhalation	Physiologic Saline Solution Inhalation	*p*-Value
Number	50	50	
Age (years ± SD)	62.46 ± 11.28	60.86 ± 11.03	0.219
Gender (m/f)			0.587
Female	7 (14%)	9 (18%)	
Male	43 (86%)	41 (82%)	
Comorbidities			0.88
Multimorbidity	15 (30%)	15 (30%)	
Neuronal	1 (2%)	1 (2%)	
Cardiovascular	21 (42%)	19 (38%)	
Hepatorenal	2 (4%)	1 (2%)	
Endocrine-metabol	14 (28%)	12 (24%)	
Pulmonal	1 (2%)	2 (4%)	
Diagnosis			1
Malignoma	48 (96%)	48 (96%)	
Inflammation	0	0	
Others	2 (4%)	2 (4%)	
Location			0.226
Oral cavity	1 (2%)	3 (6%)	
Nasopharynx	1 (2%)	0	
Oropharynx	19 (38%)	25 (50%)	
Hypopharynx	5 (10%)	6 (12%)	
Larynx	24 (48%)	16 (32%)	
Therapy			0.427
Tumor surgery	40 (80%)	43 (86%)	
Tracheostomy alone	10 (20%)	7 (14%)	
Alcohol			0.95
Never	21 (42%)	18 (36%)	
Occasionally	12 (24%)	16 (32%)	
Daily	10 (20%)	12 (24%)	
Former abuse	7 (14%)	3 (6%)	
Information missing	0	1 (2%)	
Nicotine			0.142
Non-smoker	8 (16%)	12 (24%)	
Smoker	29 (58%)	27 (54%)	
Former smoker	12 (24%)	7 (14%)	
Information missing	1 (2%)	4 (8%)	
Packyears (n ± SD)	31.37 ± 21.82	28.37 ± 19.85	0.67
Antibiotics prior to tracheostomy	1/50	1/50	1

**Table 2 jcm-10-03312-t002:** Interleukin-6 levels at Day 1, Day 3 and Day 10 after tracheostomy, Delta Day 1–Day 10: difference between the values at Day 1 and Day 10.

Variables	Liposomal Inhalation	Physiologic Saline Inhalation	*p*-Value
IL-6 Level Day 1 (pg/mL ± SD)	31,975.81 ± 41,771.89	21,093.45 ± 25,280.55	0.118
IL-6 Level Day 3	13,622.90 ± 21,538.70	15,779.50 ± 23,807.35	0.636
IL-6 Level Day 10	9491.67 ± 15,809.79	19,358.50 ± 36,130.19	0.080
Delta Day 1–Day 10	22,484.14 ± 35,172.53	1734.94 ± 26,311.94	0.001

**Table 3 jcm-10-03312-t003:** Respiratory impairment scored by endoscopy and suction maneuvers at Day 1, Day 3 and Day 10 after tracheostomy; Respiratory impairment: sum of single values of the medical scoring system, including frequency of suction maneuver, bronchoscopical assessment and assessment of mucous congestion.

Variables	Liposomal Inhalation	Physiologic Saline Inhalation	*p*-Value
Respiratory impairment Day 1	3.2 ± 1.2	3.1 ± 1.2	0.806
Respiratory impairment Day 3	2.7 ± 1.2	3.2 ± 1.0	0.016
Respiratory impairment Day 10	1.8 ± 1.6	2.4 ± 1.4	0.051
Delta Day 1–Day 10	1.4 ± 1.8	0.7 ± 1.5	0.040

**Table 4 jcm-10-03312-t004:** Blood and serum inflammatory parameter at Day 1, Day 3 and Day 10 after tracheostomy.

Variables	Liposomal Inhalation	Physiologic Saline Inhalation	*p*-Value
White blood cell count (WBCC) Day 1 (10^3^/µL)	11.19 ± 3.79	11.36 ± 3.04	0.809
WBCC Day 3	9.86 ± 3.12	10.11 ± 2.71	0.695
WBCC Day 10	8.25 ± 2.57	8.48 ± 3.22	0.698
WBCC Delta Day 1–Day 10	3.61 ± 4.38	3.51 ± 4.16	0.911
CRP Day 1 (mg/dL)	8.57 ± 5.13	13.00 ± 7.24	0.674
CRP Day 3	13.00 ± 7.24	12.06 ± 5.91	0.508
CRP Day 10	3.50 ± 3.16	4.48 ± 5.90	0.325
CRP Delta Day 1–Day 10	5.27 ± 4.88	4.11 ± 7.40	0.357

**Table 5 jcm-10-03312-t005:** Subjective overall impairment at Day 1, Day 3 and Day 10 after tracheostomy. Subjective overall impairment: sum of the visual analogue scales of the parameter coughing frequency, breathlessness, mucous congestion, color and consistency of tracheal secretion.

Variables	Liposomal Inhalation	Physiologic Saline Inhalation	*p*-Value
Subjective overall impairment Day 1	26.64 ± 8.81	23.33 ± 7.41	0.045
Subjective overall impairment Day 3	24.67 ± 8.26	24.14 ± 9.02	0.759
Subjective overall impairment Day 10	19.72 ± 9.66	23.04 ± 10.26	0.100
Delta Day 1–Day 10	6.93 ± 12.18	0.29 ± 8.00	0.002

## Data Availability

The data presented in this study are available on request from the corresponding author.

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
