# Peer review of "Liposomal Inhalation after Tracheostomy—A Randomized Controlled Trial"

_jcm, 2021, doi:10.3390/jcm10153312_

Round 1
Reviewer 1 Report
- Explain why one patient was unblinded in the control group.
- How was done the blindness between the liposomal inhalation and saline for this study? Weren't they distinguished by their containers?
- Describe more details about measuring IL-6 in the trachea and the scoring system of respiratory impairment and subjective patient's impairment. The measurement of IL-6 in the trachea is very unfamiliar to most clinicians.
- I wonder whether the CRP and WBC count interfered with patients' general or systemic conditions.
- I think the conclusion seemed to be too strong based on the result of this study.
Author Response
Explain why one patient was unblinded in the control group.
Response: The patient complaint a subjective sensation of shortness of breath. Therefore the patient was unblinded to identify if this was a potential side effect of the liposomal inhalation. Since he received an inhalation with physiological saline solution and the subjective sensation was not likely to be caused by this inhalation (and he had no drop in his peripheral oxygen saturation) the inhalation was continued according to protocol. Information on this was added to the manuscript.
How was done the blindness between the liposomal inhalation and saline for this study? Weren't they distinguished by their containers?
Response: The following is a quote from the study protocol:
“The study is carried out by 2 investigators. Investigator 1 takes over the randomization and prepares the inhalation solution according to the randomization by adding 0.9% NaCl for the control group or 3ml (24 sprays) of the LipoAersol© directly into the ready-to-use cold air nebulizer. No conclusions can be drawn about the inhalation solution used on the ready-to-use cold air nebuliser or under the nebulisation itself. If necessary, the cold air nebuliser is refilled according to the randomization by investigator 1. The preparation of the LipoAerosol© or the physiological saline solution is carried out by investigator 1 who does not carry out the further examinations according to the medical questionnaire.
Further information on this was added to the manuscript.
Describe more details about measuring IL-6 in the trachea and the scoring system of respiratory impairment and subjective patient's impairment. The measurement of IL-6 in the trachea is very unfamiliar to most clinicians.
Response: Further information especially on the measurement of IL-6 in the tracheal secretion was inserted in the M&M section of the manuscript.
I wonder whether the CRP and WBC count interfered with patients' general or systemic conditions.
Response: Usually both CRP and WBC correlate with systemic infection and in individual patients these parameter were increased postoperatively. The average values of both parameter did decrease in both groups significantly down to normal values in both groups without showing a different course between the both groups. The IL-6 values measured in the tracheal secretion did develop indepently from the systemic parameter and did not show any correlation.
I think the conclusion seemed to be too strong based on the result of this study.
Response: The conclusion has been adjusted, in particular with regard to the comment on the long-term course.
Reviewer 2 Report
This paper is well written, well structured and interesting for its practical clinical implications. Methodology is meticulous, completed by a sufficient statistical analysis. Discussion is well performed and congruent with work's aim and data already present in literature. However the sample analyzed appears to be submitted to a too short period of observation and doesn't allow to reach definitive conclusions on a longer follow up.
Author Response
This paper is well written, well structured and interesting for its practical clinical implications. Methodology is meticulous, completed by a sufficient statistical analysis. Discussion is well performed and congruent with work's aim and data already present in literature. However the sample analyzed appears to be submitted to a too short period of observation and doesn't allow to reach definitive conclusions on a longer follow up.
Response: Thank you very much for your comments on our manuscript. The conclusion has been adjusted, in particular with regard to the comment on the long-term course.